# Digital Workflow for Prosthetically Driven Implants Placement and Digital Cross Mounting: A Retrospective Case Series

Marco Tallarico [1,*], Davide Galiffi [1], Roberto Scrascia [2], Maurizio Gualandri [3], Łukasz Zadrożny [4], Marta Czajkowska [5], Santo Catapano [6], Francesco Grande [6], Edoardo Baldoni [1], Aurea Immacolata Lumbau [1], Silvio Mario Meloni [1] and Milena Pisano [1]

1   Department of Medicine, Surgery, and Pharmacy, University of Sassari, 07021 Sassari, Italy; davidegaliffi@gmail.com (D.G.); baldoni@uniss.it (E.B.); alumbau@uniss.it (A.I.L.); melonisilviomario@yahoo.it (S.M.M.); milenapisano@yahoo.it (M.P.)
2   Independent Researcher, 74121 Taranto, Italy; roberto.scrascia@gmail.com
3   Independent Researcher, 00100 Rome, Italy; mauriziogualandri@gmail.com
4   Department of Dental Propaedeutics and Prophylaxis, Medical University of Warsaw, 02-006 Warsaw, Poland; lzadrozny@wum.edu.pl
5   Department of Laryngology, Medical University of Silesia, 40-027 Katowice, Poland; mrtczajkowska@gmail.com
6   Department of Prosthodontics, University of Ferrara, Via Luigi Borsari 46, 44121 Ferrara, Italy; cts@unife.it (S.C.); francesco.grande90@gmail.com (F.G.)
*   Correspondence: me@studiomarcotallarico.it

**Abstract:** Fully digital workflow in implant dentistry is ever increasing. Treatment of partial edentulous cases is well-documented; nevertheless, complete edentulous cases are still a challenge. To present several innovations in the treatment of complete edentulous patients using digital solutions, both for implant placement and restoration delivery, was the objective of this study. It was designed as a retrospective case series study, aimed to tune further research with larger sample size, and a longer follow-up. Patients requiring complete, implant-supported restoration were asked to participate in this study. Enrolled patients were treated with four implants, immediate loading and a definitive complete arch restoration. Patients were treated using computer-assisted, template-based surgery. Multi-piece surgical templates were used to accurately place the implants, to manage the bone if needed and to make immediate loading procedure quicker and easier. After osseointegration period, definitive, extra-oral, digital impressions were taken using newly developed scan analogs, connected in the patient mouth using temporary cylinders and stabilized by means of the low-shrinkage, flowable, resin composite. Outcomes were implant and prosthesis survival rate, complications, accuracy, and patient satisfaction. Radiographic evaluation performed with a preliminary, radiopaque aluminum try-in, was used to test the accuracy of the digital impressions. Overall, 20 implants were placed in five patients. All the implants osseointegrated without complications. One impression was taken a second time due to inaccuracy of the aluminum tray-in. Finally, all of the patients were completely satisfied with both surgical and prosthetic procedures. Within the limitations of this case series, multi-piece surgical templates showed promising results improving the clinician's confidence in the case of bone reduction, post-extractive implants and immediate loading. The prosthetic template increased the trueness of the digital impression for complete edentulous patients. Finally, even if an impression was performed again, the scan-analog used for extra-oral chair-side digital impressions seemed to be a promising tool. Continuous improvements and further study are needed to confirm these preliminary results.

**Keywords:** dental implants; digital workflow; guided surgery; prosthesis

## 1. Introduction

Integrated treatment planning with dental implants is a well-established solution. Nevertheless, the achievement of the optimal implant position, based on the prosthetic

plan, is still a critical and mostly unknown consideration in implant-based surgery. An ideal prosthesis design may reduce the risk of technical and biological complications and it may allow for adequate oral hygiene maintenance [1]. Moreover, an accurate restorative-driven implant placement offers important long-term advantages, allowing for favorable aesthetics and function [2], as well as optimal occlusion and masticatory forces distribution [3].

The development digital dental equipment, including cone-beam computed tomography, intra-oral scanners, and dedicated software which allow for virtual implant planning, dramatically improve guided implant placement, making it safer, easier and affordable treatment for many [4].

There are a lot of benefits to guide the implant precisely, toward a prosthetically driven approach, such as the best possible prosthetic design, better aesthetics, optimized loading, function, and hygiene maintenance. All of this allows for long-term stability and success of the implant-supported restorations [2,5]. However, accurate impression, and precise prosthetic set-up are mandatory to properly plan the implants [6].

Although soft tissue management still represents an important issue in implant dentistry, computer-assisted template-based surgery has also changed the surgical paradigm of using extensive flaps to obtain a proper view of the surgical area, due to guided surgery has become more accurate. In this regard, flap design can be optimized to maintain or augment the keratinized soft tissue around implant-supported restorations.

Digital technologies have led to significant changes in the production process of surgical templates [7–9]. Today, several fully digital protocols for guided implant placement are available in the literature, firstly depending on methods of support, distinguished in partial and complete edentulous patients. Guided surgery in partial edentulous patients, with at least five residual teeth in two quadrants, is currently very precise [10]. On the other hand, guided surgery in complete edentulous patients is still a challenge. The main difference between them is the presence of residual teeth that act as reference point at both, surgical and prosthetic level.

The aim of the present case series study is to present a hybrid analog/digital protocol for the treatment of completely edentulous patients. Newly developed tools were tested. Multi-piece, surgical templates without metallic sleeves were used for implants placement and immediate loading procedures. Definitive digital impressions were taken using an intra oral scanner with dedicate impression tray, derived from the functionalized temporary restoration. Finally, extra-oral, definitive digital impressions were taken using novel scan analogs.

## 2. Material and Methods

This study was designed as retrospective case series evaluation, aimed to tune further studies with larger sample size and follow-up. Any healthy patients aged 18 years or older that were in need of a complete arch rehabilitation were asked to participate in this study. General exclusion criteria were adopted (Table 1).

Any patient, aged 18 years or older, with complete edentulism or a failing dentition in at least one arch, in need of a complete, implant-supported restoration, was considered eligible for the study. Patients were carefully informed about the surgical and prosthetic procedures, benefits, potential risks and complications, as well as any follow-up evaluations required for the clinical study. Before entering into the study, the patients had to sign the informed consent. The 2013 Helsinki declaration, as well as the Good Medical Practice principles, were adhered too. Medical data were anonymized so that patients cannot be identified. Due to the retrospective nature of the research, and the staging of the research (phase IV clinical trial or post-marketing surveillance trial) ethical committee approval was not requested.

All of the surgical and prosthetic procedures were performed by one expert implantologist (MT). All of the enrolled patients were treated using computer/assisted, template-based surgery with surgical templates without metallic sleeves (OneGuide kit, Osstem Implant, Osstem Global Co., Ltd., Seoul, South Korea). All of the patients received four implants

(Osstem, Osstem Implant), placed according to the manufacturer's instructions. Immediate loading was performed the same day of the surgery using a pre-fabricated metal-reinforced, temporary restoration, derived from the implant planning. Multi-piece surgical templates were used both for implant placement and immediate loading procedure.

**Table 1.** Exclusion criteria.

| Exclusion Criteria |
| --- |
| ASA 3 and 4 classification; American Society of Anesthesiologists, https://www.asahq.org (accessed on 10 July 2022) |
| Pregnant or nursing |
| Intravenous bisphosphonate therapy |
| Alcohol or drug abuse |
| Heavy smoking ($\geq$20 cigarettes/day) |
| Radiation therapy to the head or neck region within the last five years |
| History of parafunction |
| Untreated periodontitis |
| Psychiatric therapy or unrealistic expectations |
| Immunosuppressed or immunocompromised |
| Lack of opposite occluding dentition/prosthesis |
| Acute infection in the area intended for implant placement |
| Need for bone augmentation |
| Full mouth bleeding on probing [BoP] and full mouth plaque index [PI] higher than 25% |
| Allergy or adverse reactions to the restorative materials |

After osseointegration period, definitive extra-oral digital impressions were taken with a customized prosthetic template (also named prototype aesthetic try-in) and the newly developed scan analogs, screwed to conventional temporary cylinders, stabilized to the template by means of low-shrinkage, flowable, resin composite. CAD/CAM definitive complete arch restorations, made on composite and titanium, were delivered. Patients were followed for at least four months after definitive prosthesis delivery.

Outcomes were implant and prosthesis survival rate, any complications, accuracy of the digital impression, evaluated both clinically and radiographically, and patient satisfaction evaluated by means of a questionnaire delivered four months after prosthesis delivery.

- An implant was considered a failure if it presented with any mobility dictating its removal, assessed by tapping or rocking the implant head with the metallic handles of two instruments; progressive marginal bone loss or infection; and any mechanical complications rendering the implant unusable, although still mechanically stable in the bone. A prosthesis was considered a failure if, in any case, it needed to be performed again. The same operator (MT) evaluated failures;
- Any biological (pain, swelling, suppuration, etc.) and/or mechanical (screw loosening, fracture of the prosthesis, etc.) complications were evaluated and recorded by the same operator (MT);
- Accuracy was tested by means of direct vision and tactile sensation, performed by applying alternately pressure on the aluminum try-in, and then the definitive metal framework (Alternate Pressure Technique), secured without screws, to determine if any movement occurs. In addition, the one-screw test proposed by Jemt and co-workers was performed in case of doubts occurred [11]. No discrepancy of the radiopaque, aluminum try-in, secured with only one screw tightened, was observed [11]. Peri-apical radiographs were taken if needed. The same operator (MT) performed both tests using a microscope magnification (10$\times$ to 16$\times$);

- Patients' satisfaction was measured by means of a questionnaire delivered four months after prosthesis delivery by an independent outcome assessor. The following questions were asked: Are you satisfied with the function of your implant-supported prosthesis?; Are you satisfied with the aesthetic outcome of your implant supported prosthesis?; Would you undergo the same therapy again? Possible answers: yes absolutely; yes partly; not sure; not really; absolutely not. An operator not previously involved in the treatment of the patient filled out the questionnaire.

An explanatory clinical case is presented below. A 62-year-old female patient was seeking a second opinion due to the existing removable partial dentures anchored to the remaining natural elements, conducted few months ago. At the clinical and radiographic examination, a patient presented with the chief complaints of tooth mobility, gingival bleeding, pain, discomfort, poor function, and unaesthetic appearance (Figures 1 and 2).

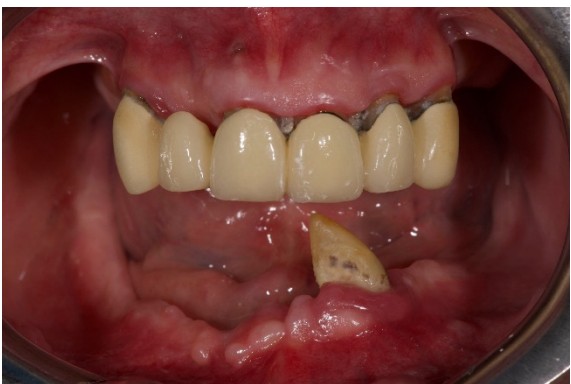

**Figure 1.** Preoperative frontal view.

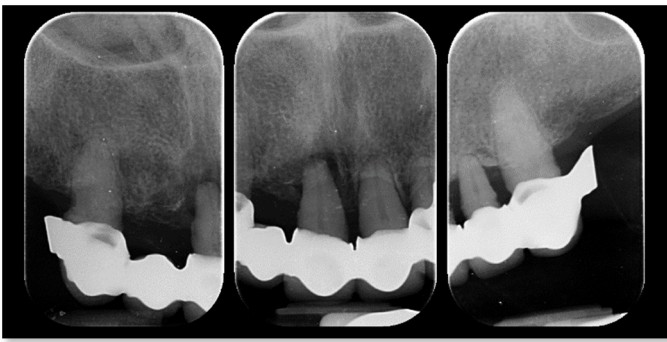

**Figure 2.** Preoperative periapical radiograph.

The final judgment was the remake of both rehabilitations. Due to economic reasons, the patient returned to previous dentist asking to resolve all the problems. However, after six years, the patient presented again seeking for anything to resolve her complaints, as they had not been addressed. The preliminary clinical and radiological analyses showed hopeless residual dentition characterized by severe chronic periodontitis, mobility, gingival bleeding, and pain in the residual maxillary teeth, and a solitary canine in the lower jaw. Aesthetic and function were also severely compromised due to patient wasn't wearing removable partial dentures for years.

Benefits and limits of several treatment options were proposed to the patient. The medical history was noncontributory; however, the patient cannot financially afford any treatment because insurance practice is still steady. At this point, the patient was asked to be enrolled in a series of dental implant courses with live surgeries. Advantages and disadvantages of this proposal have been clearly discussed with the patient and summarized in Table 2. Finally, the decision for an implant retained overdenture in the lower jaw and a fixed, screw-retained, implant supported rehabilitation in the upper jaw was

made by mutual agreement. A signed, customized, informed consent was obtained prior to commencement of the therapy.

**Table 2.** Advantages and disadvantages proposed to the patient.

| Advantages | Disadvantage |
|---|---|
| All the treatments will be free of charge | Costs for travel and medicines at her own |
| All the surgical and prosthetic procedure will be performed by the same expert clinician (M.T.) | Availability of time for the trying sessions and live surgeries |
| General rules of Good Medical Practice will be respected | Availability for follow-ups |
| Management of possible complications, including remakes will be free of charge for at least 5 years after delivery | Signed informed consent for facial pictures and video shooting |

Therapy was dived in three major steps: diagnostic (or functional), temporary and definitive rehabilitation.

Diagnostic therapy was aimed at re-educating the patient that was edentulous for several years. Immediate, diagnostic, complete removable upper and lower dentures were planned. This first step was completely analogical. Preliminary impressions were taken in alginate to pour study models. Definitive impressions were taken with customized impression tray. Centric relation and occlusal vertical dimension were recorded using Dawson's maneuver and phonetic sounds, respectively. Master models were mounted in a semi-adjustable articulator with facebow. After that, the immediate removable denture was finalized using posterior teeth with 0-degree of cuspal angulations in order to allow for mandible repositioning. The day of the first surgery, the patient underwent local anesthesia (4% articaine with adrenaline 1/100,000, Septanest, Septodont, Mataró, España). All of the residual teeth were carefully extracted, and the bone crest modeled. The immediate, diagnostic, complete removable denture were then delivered. The base and the occlusion were adjusted, and patient was followed every two weeks for four months. At each examination, occlusion was adjusted in order to find the correct mandibular position, without modifying the vertical dimension of occlusion. Four months later, the patient was pleased with the new function and aesthetic. The next step was to plan the prosthetically driven implants placement. The pre-existing, diagnostic, functionalized removable dentures were duplicated and used as customized impression trays to take new definitive impressions. Cross mounting technique was used to mount, in the dental articulator, master models and duplicates in the centric relation, and maintaining the actual occlusal vertical dimension. Finally, new temporary prostheses were fabricated with fully anatomical, occlusal surface anatomy (Figure 3). At this stage, bilateral balanced occlusion was delivered in all of the treated patients.

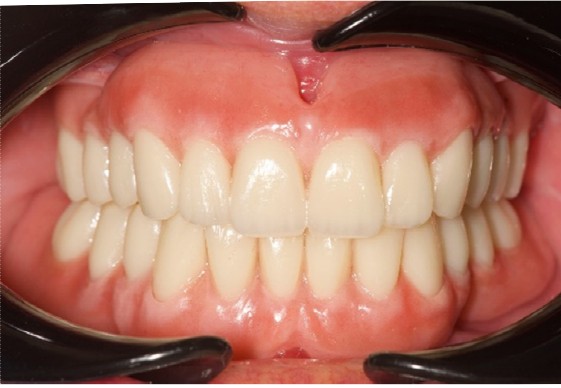

**Figure 3.** New complete removable dentures, based on the diagnostic dentures.

At this point, all of the next steps, from implant planning to definitive restorations delivery, were performed completely digitally. After the new prostheses were manufactured, the patient underwent a CBCT scan of the upper and lower jaws, according to the modified double scan technique. For the latter, tridimensional composite markers were applied on the mandibular prosthesis, and a wax bite was used to separate dental arches. After that second scan of the lower prosthesis alone was performed using an intra-oral scanner (Medit i500). In the lower jaw, two straight implants were planned (Figure 4) and placed (Figures 5 and 6) to retain a classical overdenture.

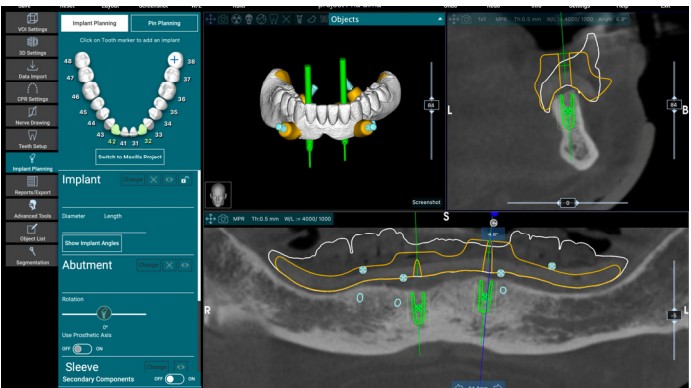

**Figure 4.** Virtual implant planning for two-implant-retained, mandibular overdenture.

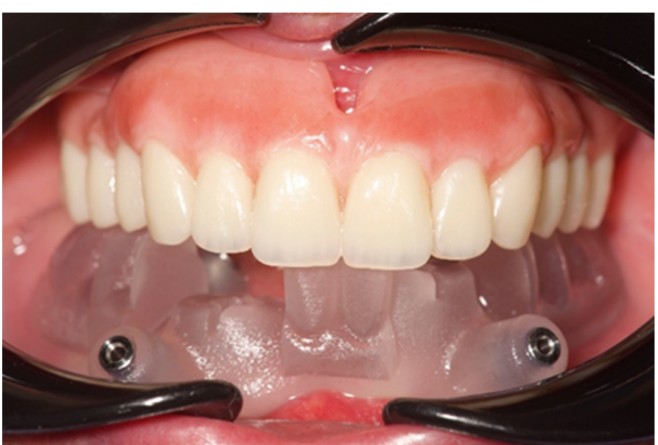

**Figure 5.** Surgical template without metallic sleeves.

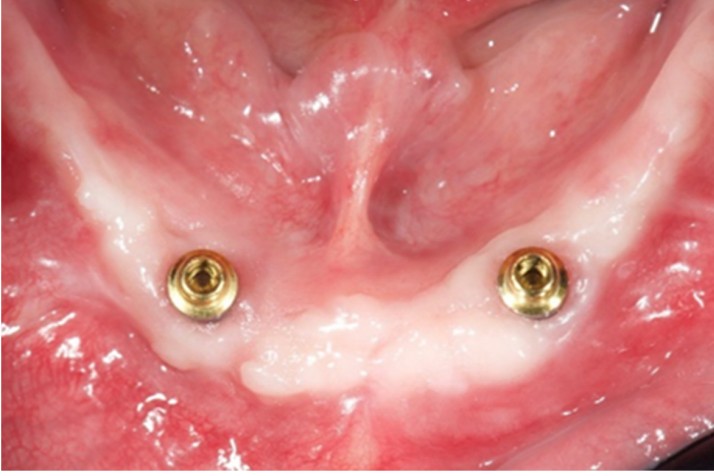

**Figure 6.** Placed implants with OT Equators (Rhein'83, Bologna, Italy).

In the upper jaw, four implants were planned according to the All-on-4 protocol (Figure 7), to support a fixed, screw-retained, dental prosthesis. In addition, a three-to-four mm of bone maxillary reduction was also planned.

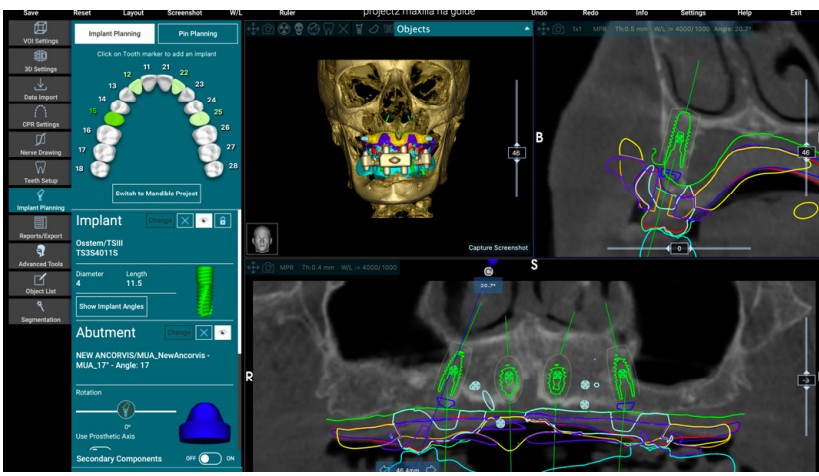

**Figure 7.** Virtual implant planning for maxillary All-on-4.

To allow for an accurate implant placement and bone reduction, a specially designed, multi-piece surgical template without metallic sleeves was printed. This template consisted of three portions (Figure 8).

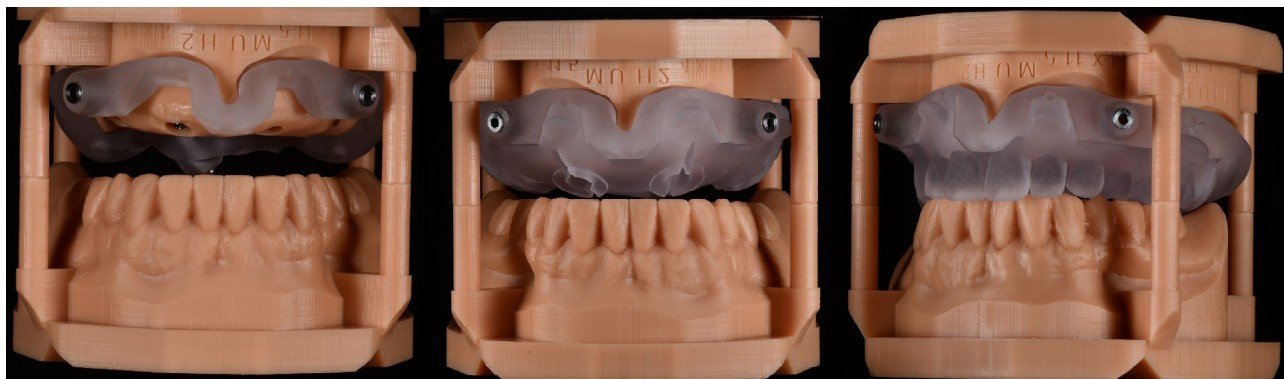

**Figure 8.** Multi-piece surgical template. From left to right: base-, implant-, and index-portion.

The base portion allows for template stabilization, bone reduction, and temporary restoration delivery; the implant portion allows for guided implant placement; and the index portion allows for template stabilization in centric relation, before surgery.

All of the implants were placed using guided surgery, but in two different surgical sections, two months apart. In both jaws, anesthesia (4% articaine with adrenaline 1/100,000, Septanest, Septodont, Mataró, España) was made in the buccal area and thought the holes of the surgical templates, about 15 min before surgery. After that, a muco-periosteal flap was done.

Mandibular implants were planned and placed parallel, and the OT Equator attachments (Rhein'83, Bologna, Italy) were immediately screwed onto the implants. The flap was sutured and the definitive overdenture were connected to the attachment systems chairside. One week later, at the sutural removal, the implants were loaded. In the upper jaw, a metal-reinforced, temporary restoration, based on the prosthetic set-up, was fabricated before surgery (Figure 9).

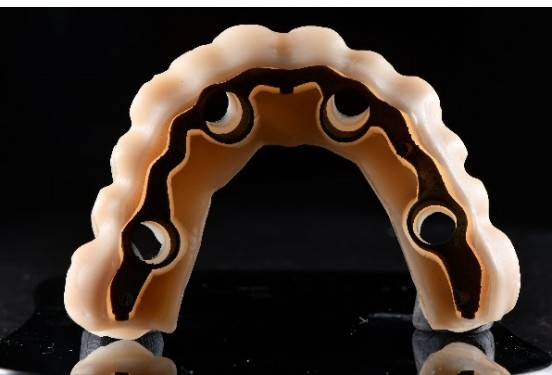

**Figure 9.** Metal-reinforced temporary restoration.

The day of the surgery, after flap elevation, bone reduction was performed using the base portion of the surgical template as a reference (Figure 10). Then, two straight anterior implants and two tilted posterior implants were placed, according to the All-on-4 protocol, using the implant portion, without metallic sleeves (Figure 11).

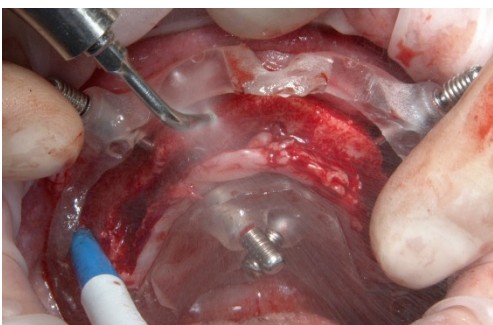

**Figure 10.** Base-portion in the patient's mouth used as reference guide during the bone reduction.

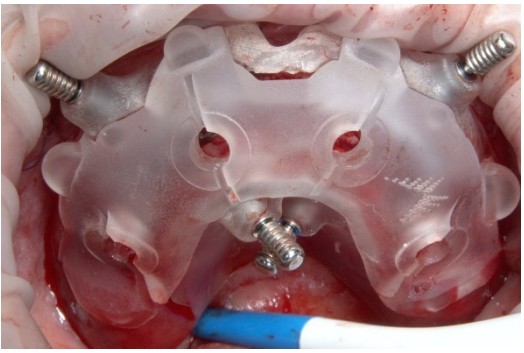

**Figure 11.** Base-portion plus implant-portion.

Multi abutment units were screwed onto the implants and the flap was sutured. A fixed, screw-retained, temporary restoration, without cantilever, was delivered. The temporary restoration, based on the maxillary set-up, was milled in PMMA (Polymethyl methacrylate) and refined before delivery. The temporary restoration was rebased in the patient mouth, using the base-portion of the multi-piece surgical template as reference for the occlusal vertical dimension and centric relation. The patient was checked every 2 weeks (Figures 12–14).

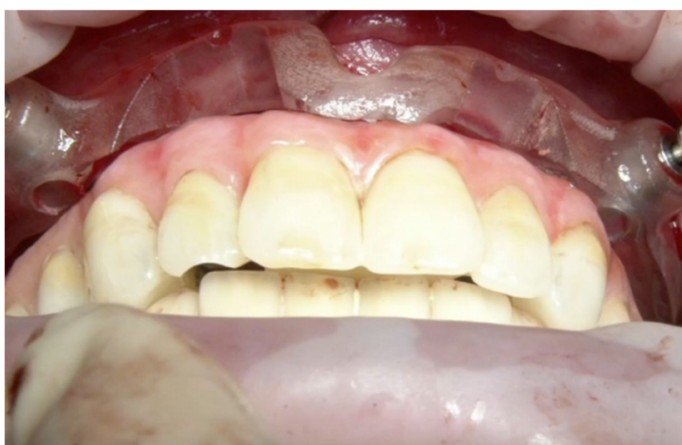

**Figure 12.** Base-portion used to guide for the immediate loading. It allows for the temporary restoration to stop at the correct vertical dimension of occlusion and centric relation.

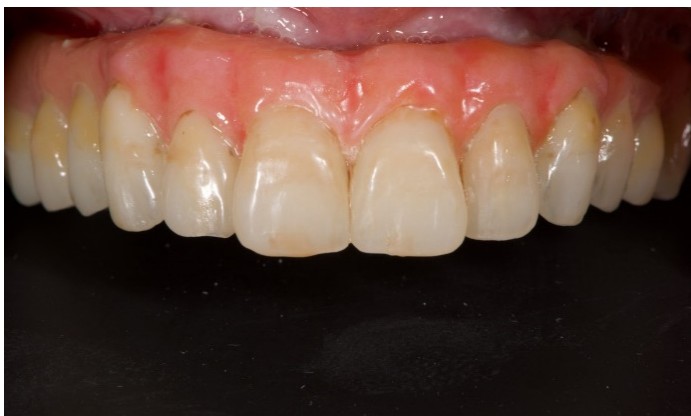

**Figure 13.** Screw-retained, temporary restoration after finishing.

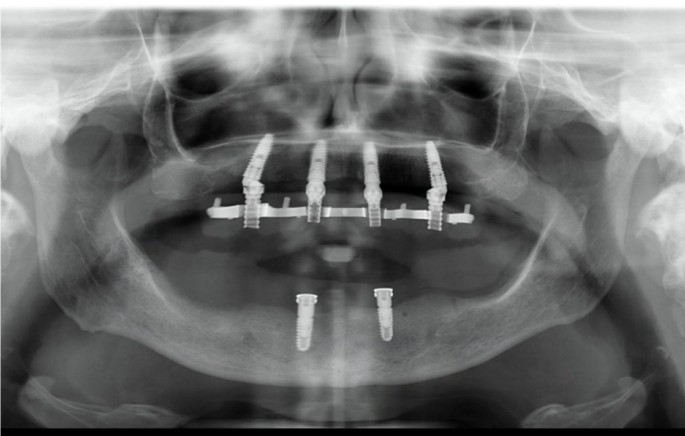

**Figure 14.** Ortopantomograph taken after implant placement.

Four months later, a preliminary digital impression of the functionalized temporary restoration was firstly taken using scan analog for multi-abutment (Figures 15–17). Then, the temporary restoration was unscrewed, and the conventional scan analog connected. A second digital impression of the implant position was taken.

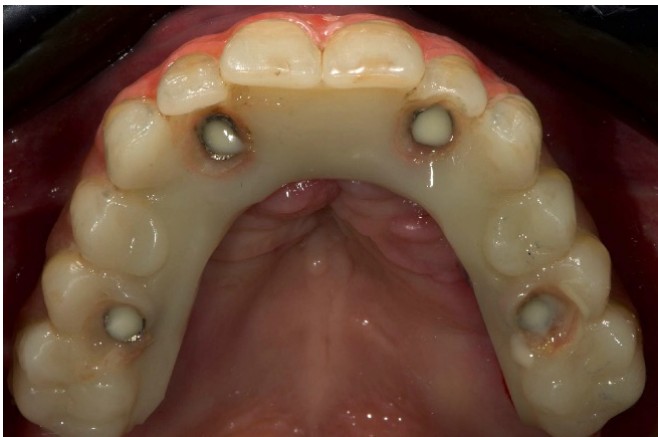

**Figure 15.** Temporary restoration, occlusal view.

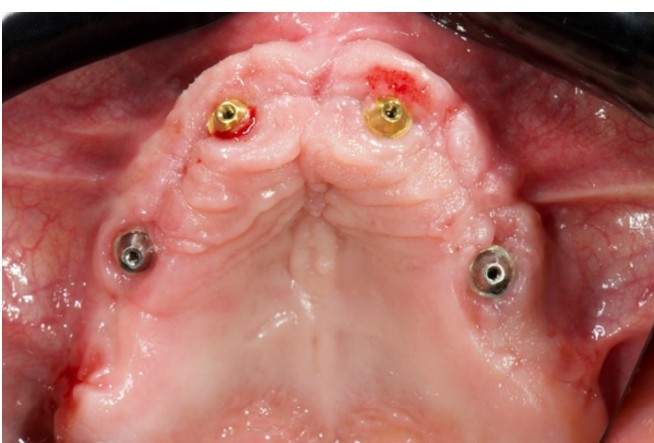

**Figure 16.** Implants with multi abutments in place: occlusal view.

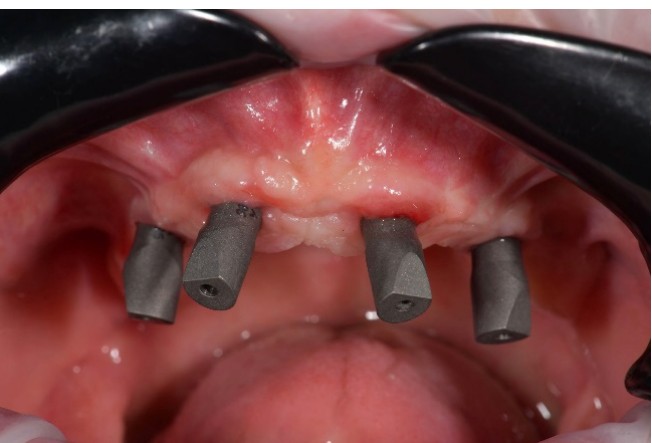

**Figure 17.** Scan abutments for multi abutments in place.

Both impressions were aligned using the palatal as a reference point (Figure 18). At this point, an aesthetic try-in was printed, and then tested in the patient mouth and used as definitive impression tray (Figure 19). After aesthetics and function were tested, the aesthetic try-in was connected to the implants using conventional temporary abutments (Figure 20).

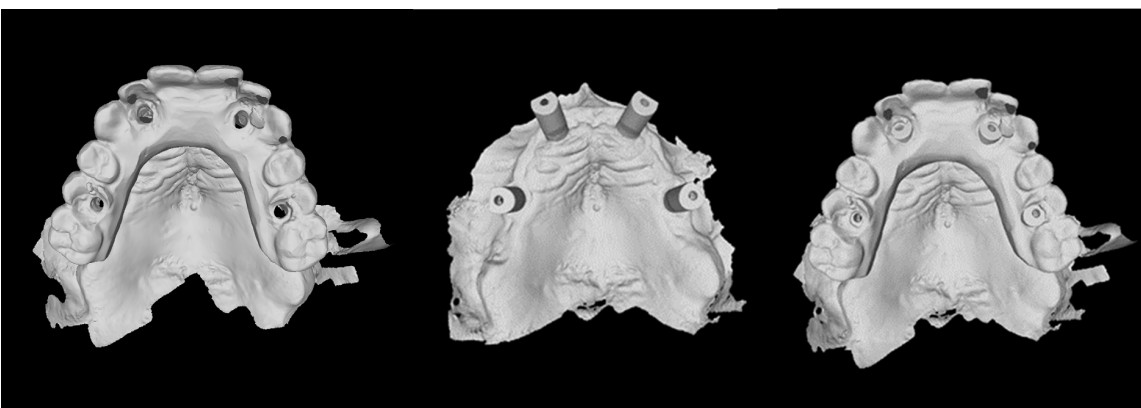

**Figure 18.** From left to tight: digital impression of the temporary restoration; matching between the two digital impressions; digital impression of the scan abutments in place.

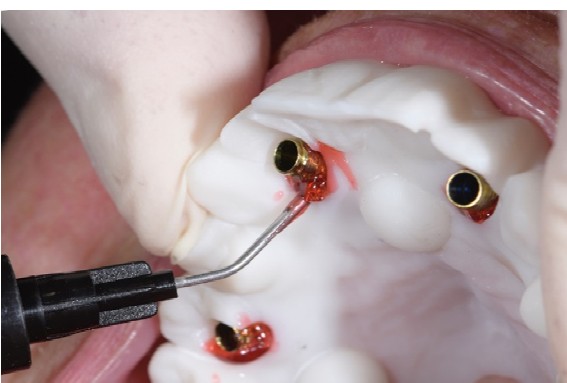

**Figure 19.** Prototype aesthetic try-in, used for the aesthetic and functional tests.

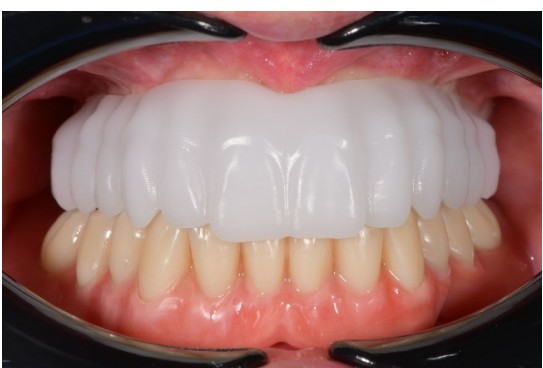

**Figure 20.** Temporary abutments connected to the aesthetic try-in.

After that, the aesthetic try-in was unscrewed. Four newly designed scan analogs were connected to the temporary abutments and scanned out of the patient's mouth, using a conventional extra-oral scanner (Figure 21). Before the definitive CAD/CAM framework was fabricated, a radiopaque, aluminium framework was tested clinically and radiographically. After that, cross mounting of the definitive restoration (implant position) with the functionalized temporary restoration, as well as the original plan, was made. At the next appointment, the titanium, CAD/CAM framework was tested in the patient's mouth. Finally, a fixed, screw-retained, implant supported restoration made in titanium, with composite as veneering material was delivered (Figures 22–24). Composite was chosen due to their resilience, and shock absorption potential [12]. Lingualized occlusion was designed for all of the definitive restorations, using anatomic teeth for the maxillary denture and modified non-anatomic or semi-anatomic teeth for the mandibular dentures.

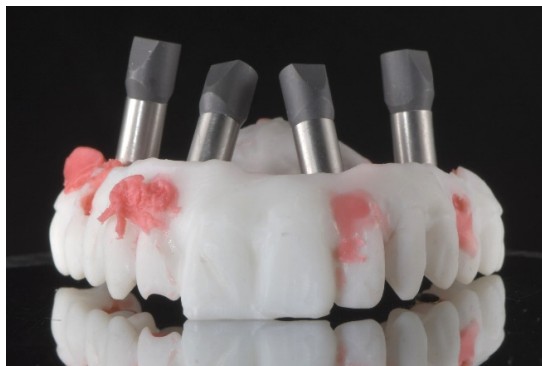

**Figure 21.** Scan analogs connected to the temporary abutments.

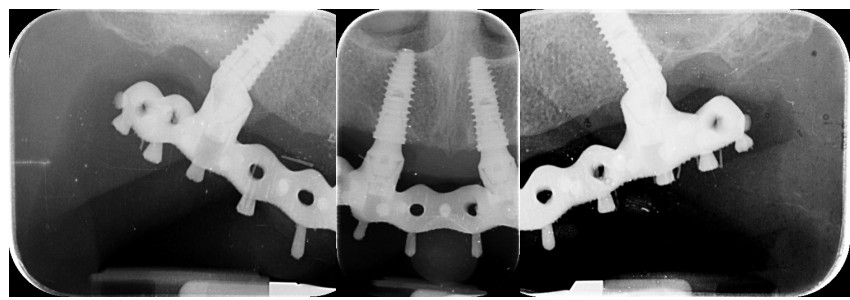

**Figure 22.** Periapical radiographs at prosthesis delivery.

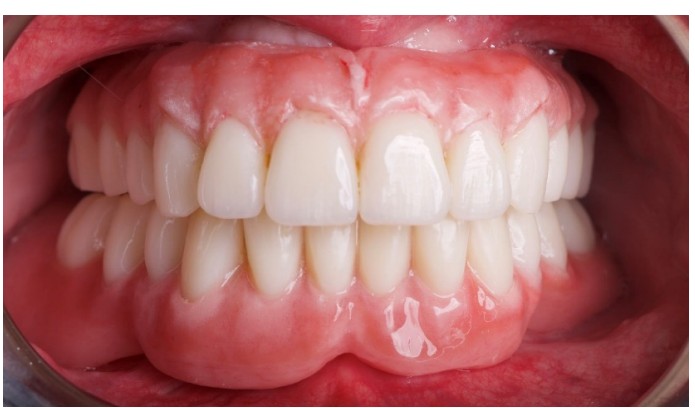

**Figure 23.** Clinical view of the maxillary definitive prosthesis one year after its delivery.

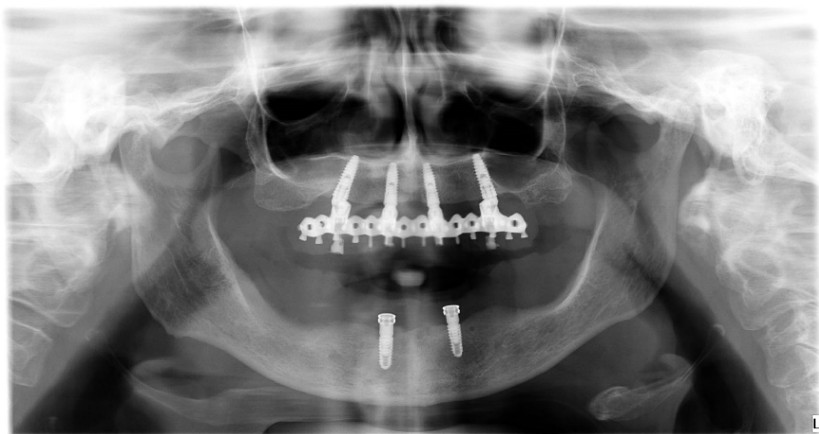

**Figure 24.** Orthopantomograph one year after prosthesis delivery.

## 3. Results

All of the implants osseointegrated successfully, and all the definitive prostheses were in function at the end of this preliminary report, resulting in an implant and prosthesis survival rate of 100%. No biological and technical complications were experienced. However, one out of five impressions was performed again due to inaccuracy of the aluminum tray-in. In this case, the aluminum tray-in was separated and the two portions were fixed using a dedicate light curing resin for implants (EZ-Pattern LC), polymerized under UV/VIS light 350–480 nm. Furthermore, a plaster impression was taken. No mechanical complications were experienced after definitive restorations delivery. All of the patients were completely satisfied with both function and aesthetic outcomes, and they were willing to repeat the procedure again.

## 4. Discussion

This research aimed to present several innovations in the digital approach for the fully digital approach and treatment of edentulous patients. However, the proposed workflow started with analog procedure to fabricate and delivery a novel complete removable denture, according to the functional and aesthetic demands of the patients. It is the authors opinion that, for this first step, analog procedures are even better. After that, the first innovation was the newly developed surgical templates without metallic sleeves to guide implants placement, designed with two-to-three assembled parts. Surgical templates without metallic sleeves have already been used in the last years, showing equal or better accuracy comparing with conventional templates [13,14]. With the limitations of the present research, multi-piece, surgical templates may allow for safe implant placement and immediate loading delivery, even in cases where bone reduction is needed. However, this concept is not new in dentistry. Multi-piece (reassembled) radiographic guides were already proposed in the literature to allow for a prosthetically-driven approach, even in case of immediate post-extractive implants [15–17]. The main benefit was to visualize the complete prosthetic set-up, and at the same time, to maintain the teeth in the patient's mouth, up to the surgical session. On the contrary, the main limitation was that the surgical template could not be tried before surgery. Polizzi and Cantoni applied this concept using a two-piece radiographic guide, tricking the software in the case teeth extraction was needed at the time of implant placement [15,16]. Today, computer-aided design software is improved, allowing for virtual wax-up in several clinical situations, including post-extractive implants. In the present research, the concept of multi-piece tools is moved to the surgical templates. Once the implants were planned in the properly, prosthetically driven position, the multi-piece surgical template is designed. The number of reassemble parts depends by the needed surgical procedures, including bone reduction and/or post-extractive implants. Following this protocol, the surgical template is completely customized according to the surgical needs.

The second innovation was to present a prosthetically driven impression tray, aimed at recording an accurate digital impression for complete arch restoration. Tallarico and co-workers, in the 2017 and 2018, presented a couple of case reports, showing the same prosthetic template [8,18]. A few years later, the same author published an in vitro study showed that the prosthetic-based impression template significantly improved the trueness and precision of complete edentulous arches rehabilitated with four or six implants, making the complete arch digital impression more predictable [19]. Similar results were obtained in a couple of case reports by Venezia and co-authors [7,20]. This approach not only permits the transfer of the inter-maxillary and occlusal relationship, but also allows for the improvement of the overall accuracy of the digital impression. It is well known that the accuracy of complete arch impressions is still challenging for intra-oral scanner devices [21]. On the other hand, limited clinical differences were found comparing digital and conventional impression taken to rehabilitate implant-supported, partial restorations [22]. For the latter, the primary purpose of the prosthetically driven impression tray is to improve the accuracy of digital complete arch impressions, adding several reference points between the scan

abutments. This concept is almost similar to taking a partial digital impression. In addition, the design of the prosthetically driven impression tray was based on the temporary restoration, functionalized in the patient mouth. Therefore, a cross-mounting technique can be performed to fabricate a perfect, aesthetic and functional copy, of the temporary restoration. Finally, aesthetics and function may be tested in the case appointment, making it easier to deliver the final restoration.

In the present study, extra-oral definitive digital impressions were taken by using a newly developed scan analogs, in combination with conventional temporary cylinders, connected to the implants in the patient's mouth. Extra-oral digital impression was already proposed by the same research group [18]. The main benefit was to overcome the drawbacks of the digital impression, such as accuracy of long-span restorations on multiple implants, difficult bite registration of complete edentulous arches, and need for a learning curve [23–25]. In addition, it is well known that the accuracy of extra-oral scanner is preferable, compared with intra-oral devices, for full-arch scanning [26]. On the other hand, the risks could be the distortion of the implant position, due to a disassembly of the scan-analog/temporary cylinder complex, from the prosthetic template. In the present research, only one impression was performed again due to inaccuracy of the aluminum try-in. It must be said that the failed impression was the first. The present research was designed as a case series study aimed to develop novel tools that may facilitated the treatment of complete edentulous patients, using digital technology. Moreover, the inaccuracy matched the scan analog. Library was custom-made, so it is possible that further improvements may eliminate this error. In fact, all the following four impressions were accurate, without misalignments.

Although the main object of this study is to present several digital innovations, the analog phase is still needed in the initial ashes of the treatment. In the present study, all of the patients were initially rehabilitated with complete removable dentures, delivered accordingly to the functional and aesthetic needs of the patients. This allowed us to plan the implants in the correct prosthetically driven positions. Moreover, in case the patients presented dysfunctional occlusion, initial rehabilitation involves a phase with posterior teeth with 0-degree of cuspal angulations. This allows for the repositioning of the mandibular condyle.

However, great progress has been made in digital dentistry in the last years. Nevertheless, literature is still lacking of consistent studies [27,28]. The main limitations of the present study are the small sample size, and the short follow-up period. Nevertheless, this research was design as case series study aimed at evaluating the feasibility of using multi-piece and multi-function surgical template, the prosthetic template, as well as the scan analog. These preliminary cases allowed us to obtain encouraging results. All of the patients were fully satisfied, and the overall functional and esthetic results met the criteria of the good practice in dentistry. Moreover, the proposed fully digital workflow could lead to a potential reduction in the time and costs and related problems for the overall treatment of edentulous patients [29,30]. Based on this preliminary report, further studies must be designed to test the proposed hybrid analog/digital work-flow, including the newly developed digital innovations.

## 5. Conclusions

Within the limitations of the present case series study, the multi-piece surgical template showed promising results that may improve the surgical confidence in case of bone reduction, post-extractive implants, and immediate loading. The prosthetic template may increase the trueness of the digital impression for complete edentulous patient. Finally, the scan analog used for extra-oral chair-side digital impression seems to be a promising tool. Further clinical studies are needed to confirm these preliminary results.

**Author Contributions:** Conceptualization, M.T. and S.M.M.; methodology, M.P.; software, R.S.; formal analysis, M.C.; investigation, M.T., D.G. and M.G.; data curation, Ł.Z.; writing—original draft preparation, M.T. and Ł.Z.; writing—review and editing, F.G.; supervision, S.C.; funding acquisition, E.B. and A.I.L. All authors have read and agreed to the published version of the manuscript.

**Funding:** This research received no external funding and the APC was funded by The University of Sassari.

**Institutional Review Board Statement:** The study was conducted in accordance with the Declaration of Helsinki, Due to the retrospective nature of the research, and the staging of the research (phase IV clinical trial or post-marketing surveillance trial) ethical committee approval was not requested.

**Informed Consent Statement:** Informed consent was obtained from all subjects involved in the study.

**Data Availability Statement:** Not applicable.

**Acknowledgments:** The authors want to thanks the New Ancorvis Srl (Via dell'Industria, 15, 40012 Bargellino, Calderara di Reno BO), Rhein'83 Srl (Via Emilio Zago, 10, 40128 Bologna BO), Micerium S.p.A. (Via Guglielmo Marconi, 83, 16036 Avegno GE), and Merz Dental GmbH (Kieferweg 1, 24321 Lütjenburg, Germany) that supported this research donating most of the used materials and products. The authors want also to thank Franco Sanseverino, for his precious support.

**Conflicts of Interest:** Some companies (New Ancorvis Srl, Rhein'83 Srl, Micerium S.p.A., and Merz Dental GmbH) partially supported this trial donating most of the used surgical and prosthetic materials used in the present investigation. Nevertheless, all data are the property of the authors, and by no means did the companies interfere with either the conduct of the trial or publication of its results. Lastly, all the authors declare that all the procedures were performed free of charge, and they have no any competing financial interests or personal relationships that could have appeared to influence the work reported in this paper.

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
