# Peer review of "Digital Workflow for Prosthetically Driven Implants Placement and Digital Cross Mounting: A Retrospective Case Series"

_prosthesis, doi:10.3390/prosthesis4030029_

Round 1
Reviewer 1 Report
Title: I believe it will be better to add the word 'retrospective' in the title which should ideally read as 'retrospective case series'.
The authors have mentioned that an ethical approval was not sought given the retrospective nature of the study. I believe that a waiver of consent could have been applied for from the local ethics committee.
Table 1: I think it will be better to rephrase 'parafunctional activity' to 'history of parafunction' as it is hard to diagnose parafunction when minimal or no teeth are present like the case presented.
Line 196: It is mentioned that 0 degree teeth were used for immediate dentures. What was the reason for choosing this occlusal scheme? Was this scheme used for all the 5 patients in this study? Were non-anatomic teeth used for definitive removable dentures as well?
Please provide more details on the occlusal scheme used for the final restorations on implants.
Was there any specific reason for using composite resin and titanium for the final implant supported fixed prosthesis?
Lines 306-307 - How was the aesthetic try-in fabricated? Was it a printed or milled replica of the temporary prosthesis? Please provide more details.
The abstract and the manuscript mention the term 'aluminium try-in'. Its hard to tell from reading the case report that how and when it was done? There is hardly any mention of this term in the case report. Please add more details to explain better.
Please proofread the manuscript again as there are a lot of grammatical and typographical errors.
Author Response
Dear reviewer, thanks a lot for your valuable comments.
Please check my step by step revision.
Title: I believe it will be better to add the word 'retrospective' in the title which should ideally read as 'retrospective case series'.
- Thanks a lot. "Retrospective" has been added in the title.
The authors have mentioned that an ethical approval was not sought given the retrospective nature of the study. I believe that a waiver of consent could have been applied for from the local ethics committee.
- Dear reviewer, I partially agree with you. In Italy, ethical approval for retrospective study is not mandatory. My Ethical committee refused to gave approval for retrospective case series study. Moreover, I checked in Prosthesis Journal and I found some manuscript with same issue. Honestly, I agree that for retrospective case series studies EC approval is not need. These are observational studies of phase IV,. This means after commercial distribution of the sued materials. These informations are reported in the manuscript. Thanks.
- "Before entering in the study, the patients had to sign the informed consent. The 2013 Helsinki declaration, as well as, the Good Medical Practice principles, were adhered too. Medical data were anonymized so that patients cannot be identified. Due to the retrospective nature of the research, and the staging of the research (phase IV clinical trial or post-marketing surveillance trial) ethical committee approval was not requested."
Table 1: I think it will be better to rephrase 'parafunctional activity' to 'history of parafunction' as it is hard to diagnose parafunction when minimal or no teeth are present like the case presented.
- Thanks for your comments. It has been rephrased.
Line 196: It is mentioned that 0 degree teeth were used for immediate dentures. What was the reason for choosing this occlusal scheme? Was this scheme used for all the 5 patients in this study? Were non-anatomic teeth used for definitive removable dentures as well?
- This point has been clarified. Thank you.
- "…posterior teeth with 0-degree of cuspal angulations in order to allows mandible repositioning"
- "Finally, new temporary prostheses were fabricated with fully anatomical, occlusal surface anatomy (Figure 3). At this stage, bilateral balanced occlusion was delivered in all the treated patients."
- "Although the main object of this study is to present several digital innovation, analog phase is still needed in the initial ashes of the treatment. In the present study, all the patients were initially rehabilitated with complete removable denture, delivered accordingly to the functional and esthetic needs of the patients. This allow to plan the implants in the correct prosthetically driven position. Moreover, in case the patients presented dysfunctional occlusion, initial rehabilitation involves a phase with posterior teeth with 0-degree of cuspal angulations. This allows repositioning of the mandibular condyle."
Please provide more details on the occlusal scheme used for the final restorations on implants.
- These informations has been added.
- "Lingualized occlusion was choose for all the definitive restorations, using anatomic teeth for the maxillary denture and modified non-anatomic or semi anatomic teeth for the mandibular dentures."
Was there any specific reason for using composite resin and titanium for the final implant supported fixed prosthesis?
- The reason has been added also with a new reference. "Composite was chosen due to their resilience, and shock absorption potential (12).".
Lines 306-307 - How was the aesthetic try-in fabricated? Was it a printed or milled replica of the temporary prosthesis? Please provide more details.
- Thanks, this point has added. "aesthetic try-in was printed, and then…".
The abstract and the manuscript mention the term 'aluminium try-in'. Its hard to tell from reading the case report that how and when it was done? There is hardly any mention of this term in the case report. Please add more details to explain better.
- This point has been explained well in all the text. Thanks.
Please proofread the manuscript again as there are a lotof grammatical and typographical errors.
- The manuscript has been proofread. Thanks.
Reviewer 2 Report
This research is under the scope of this journal; the topic is relevant for readers, and this research deals with potentially significant knowledge to the field.
- There are many mistakes in the references section and in the text
- Limitations?
Conclusions were not totally supported by the data showed - Figure legends: Bad descriptions
Author Response
Dear reviewer, thanks a lot for your comments.
This research is under the scope of this journal; the topic is relevant for readers, and this research deals with potentially significant knowledge to the field.
There are many mistakes in the references section and in the text
- References and text have been checked.
Limitations?
- Limitations have been added. Thanks.
Conclusions were not totally supported by the data showed
- Conclusions have been reviewed, accordingly.
Figure legends: Bad descriptions
- Legends have been revisited.
Thanks